PLOS **Neglected Tropical Diseases**

# Hematological diseases-related mucormycosis: A retrospective single center study

**Yu Cui[1], Rui Zhao[1], Ruihua Mi[1], Lin Chen[1], Lin Wang[1], Dongbei Li[2], Cheng Cheng[1], Mengjuan Li[1], Xudong Wei**[1]*

**1** Department of Hematopathy, The Affiliated Cancer Hospital of Zhengzhou University & Henan Cancer Hospital, Zhengzhou, China, **2** Center Laboratory, The Affiliated Cancer Hospital of Zhengzhou University & Henan Cancer Hospital, Zhengzhou, China

* xudongwei@zzu.edu.cn

## Abstract

### Background and aim

Mucormycosis is a life-threatening invasive fungal infection. This study aimed to analyze the clinical characteristics of patients with hematologic malignances complicated with mucormycosis.

### Methods

This retrospective study investigated the clinical characteristics, epidemiological features, treatment, and prognosis of 46 patients with hematological diseases and Mucor infection as indicated by mNGS from August 28, 2020 to September 11, 2023. Metagenomic next-generation sequencing (mNGS) refers to the application of high-throughput sequencing technology for the comprehensive analysis of nucleic acid content in patient samples, facilitating the detection and characterization of microbial DNA and/or RNA, and then comparing and analyzing the results with an information database to determine the types of pathogenic microorganisms present in the sample.

### Results

The median age of admission for the included patients was 49 years (9–78). Multivariate analysis identified age over 60 years (p = 0.006 < 0.05), high-dose corticosteroids (p = 0.001 < 0.05), neutropenia lasting more than 10 days (p = 0.041 < 0.05), and two or more Mucor infections (p = 0.004 < 0.05) were independent risk factors for OS in patients with hematological diseases. Moreover, differences between groups were analyzed using the Fisher exact probability method, and no significant difference was observed in the efficacy of various types of antifungal therapies.

**Data availability statement:** All relevant data are within the manuscript and its Supporting Information files.

**Funding:** This work was supported by the National Natural Science Foundation of China(NO.82170151 to X.W.), China International Medical Foundation (NO.Z-2018-35-2003 to M.L.), Natural Science Foundation of Henan Province (NO.232300420238 to R.M.), Henan provincial Medical Science and Technology Research Project (NO.LHGJ20210185 to R.M.), Natural Science Foundation of Henan Province (NO.242300421507 to D.L.), Henan Province Outstanding Young and Middle-aged Talents Cultivation Project for Health Science and Technology Innovation (NO.JQRC2023009 to R.M.). The funders had no role in study design, data collection and analysis, decision to publish, or preparation of the manuscript.

**Competing interests:** The authors have declared that no competing interests exist.

## Conclusion

Patients with hematologic malignancies benefit greatly from early diagnosis and treatment when suspected of Mucor infection. mNGS is an important supplementary method for early diagnosis of Mucor infection. Moderated use of corticosteroids, reducing the duration of neutropenia, and enhancing autologous immune function are important measures to reduce patient mortality rate.

## Author summary

Mucormycosis is an infrequent yet highly lethal invasive fungal infection. Prompt diagnosis and intervention are crucial. The clinical presentations of mucormycosis in patients with concurrent hematological malignancies often lack specificity, rendering early clinical detection challenging. In recent years, metagenomic next-generation sequencing (mNGS) has emerged as a novel diagnostic technology. This method facilitates the identification of non-cultivable pathogenic microorganisms by employing high-throughput sequencing technology (Next-Generation Sequencing, NGS) to sequence nucleic acid molecules from microorganisms present in environmental or biological samples without discrimination or bias. The sequencing data are subsequently compared and analyzed against a comprehensive microbial sequence database to perform qualitative or quantitative assessments of the microorganisms present in the samples. Notably, mNGS does not require specific amplification and is characterized by its rapidity, sensitivity, and accuracy, making it a valuable adjunct to traditional diagnostic methods such as culture. In this study, we conducted a retrospective analysis of the clinical characteristics, epidemiological features, diagnosis, treatment, and prognosis of 46 patients with suspected hematological malignancies complicated by mucormycosis. Our findings underscore the critical importance of promptly conducting relevant examinations when mucormycosis infection is suspected in patients with hematological malignancies. mNGS emerges as a potentially valuable adjunctive tool for early diagnosis, particularly in cases where histopathological examination or culture is not feasible. We highlight the necessity of timely diagnosis and intervention, judicious use of corticosteroids, regulation of neutrophil counts to mitigate risk factors, and enhancement of the patient's immune function as pivotal strategies to reduce mortality rates in patients with hematological malignancies complicated by mucormycosis.

## Introduction

Mucor is a thermotolerant fungus widely present in organic matrices, with sporangium serving as the primary mode of transmission [1]. Mucormycosis, an infection caused by Mucor fungi, exhibits rapid progression and high mortality [2]. The diagnosis of mucormycosis in its early stages can be difficult due to the fact that histopathology and routine

culture are currently the standard diagnosis procedures. However, biopsy is often not feasible for patients with hematological diseases, and traditional pathogen culture's high false-negative rate and time-consuming nature render it suboptimal for early diagnoses [3,4]. mNGS refers to the application of high-throughput sequencing technology for the comprehensive analysis of nucleic acid content in patient samples, facilitating the detection and characterization of microbial DNA and/or RNA, and then comparing and analyzing the results with an information database to determine the types of pathogenic microorganisms present in the sample [5]. mNGS offers an unbiased approach to detecting a broad spectrum of infection types. It has emerged as a rapid, sensitive, and accurate technique for pathogen identification, serving as a valuable complement to conventional methods, such as pathogen culture, particularly in patients undergoing hematopoietic stem cell therapy [6].

It is now understood that hematological malignancies, transplantation (hematopoietic stem cell transplantation, solid organ transplantation), neutropenia, diabetes, and long-term corticosteroid usage are risk factors for mucormycosis [7,8]. Based on organ involvement, mucormycosis can be categorized into rhinocerebral, pulmonary, gastrointestinal, cutaneous, and disseminated types. In patients with hematological diseases and transplant, rhinocerebral involvement is most common [9], followed by pulmonary involvement [10,11]. Current evidence suggests that the mortality rate of mucormycosis patients with hematological malignant tumors is more than 50%. In recent years, the COVID-19 pandemic has led to immune disorders and increased use of steroids, and the incidence of mucormycosis has soared [12]. To enhance the treatment status of the population and decrease mortality rates, we conducted a retrospective analysis of the epidemiology, clinical characteristics, diagnosis, and treatment, and prognosis of 46 patients suspected to have hematological diseases complicated by Mucor infection.

## Methods

### Ethics statement

This study was approved by the Institutional Ethics Committee of Henan Cancer Hospital (China) (approval number: 2022-548-001). All patients provided verbal informed consent for the use of their data in this study.

### Patients

From August 28, 2020, to September 11, 2023, forty-six inpatients at the affiliated Cancer Hospital of Zhengzhou University with suspected mucormycosis and hematological diseases were included. The suspicion date for invasive fungal infection (IFD) was designated as $D_0$, with all patients undergoing mNGS within 20 days. Adhering to the definitions provided by the European Organization for Cancer Research and Treatment (EORTC) and the Fungal Research Group Education and Research Community (MSGERC) for invasive mycosis, cases were classified as "Proven" or "Probable" [13]. Neutrophil counts, imaging (CT, MRI), microbiological data (culture, mNGS), histopathology, details of clinical treatment, and treatment outcomes were collected for retrospective analysis. All patients underwent physical examinations, chest X-rays, computed tomography (CT) scans, or a combination of these diagnostic measures; all patients also underwent culture and high-throughput sequencing. Furthermore, all patients exhibited host factors and clinical manifestations indicative of Mucor infection.

### Diagnostic criteria of mucormycosis

According to the definition of invasive mycosis by the EORTC and MSGERC, the classification of cases was conducted. The "Proven" category (n = 9) required positive findings in culture and/or histopathology, while the "Probable" category (n = 37) was diagnosed based on assessments by imaging experts and clinicians at our hospital who evaluated the diagnosis according to the definitions.

### mNGS and analysis

Samples from peripheral blood or infected sites were promptly collected and dispatched to the laboratory within a 12-hour timeframe for nucleic acid extraction. The extracted DNA underwent library construction. The DNA fragments were then

sized to 200~300 bp through DNA enzyme cleavage, followed by polymerase chain reaction (PCR) amplification (2012B; Genskey) for library construction and quality control. Subsequently, 2~3 groups of quantified circular single-strand DNA were introduced to generate DNA nanospheres. These DNA nanospheres were applied to the sequencing chip.

The low-quality original reads were filtered. Subsequently, bowtie2 (v2.3.4.3) was employed to eliminate reading segments that aligned with the human reference genome GRCh38, generating high-quality sequences. The remaining data was then compared with a microbial database. The mapped reads, referred to as "themappedreads", were classified using the National Biotechnology Information Center Genome Database.

### Definition

Neutropenia was characterized by a peripheral blood neutrophil count less than $0.5 \times 10^9$/ L. Persistent neutropenia was determined when neutropenia persisted at a level of $0.5 \times 10^9$/ L or lower for three consecutive days after diagnosis. High-dose corticosteroids were identified as an oral or intravenous prednisone dose exceeding 500mg within one month before the onset of Mucor infection. Combination therapy was defined as the use of any combination involving amphotericin B (AmB), triazole, and Echinocandins either as an initial treatment or as salvage therapy.

### Statistical analysis

Statistical analysis was conducted using SPSS 27.0. The Kaplan-Meier method was employed to generate survival curves, and the Log-Rank test was utilized to assess differences in survival. Univariate and multivariate Cox regression analyses were performed to identify independent factors influencing mortality. Differences between groups were analyzed using the Fisher exact probability method, and a P-value < 0.05 was statistically significant.

### Result

The median age of admission for the included patients was 49 years (9–78), with 69.6% (32/46) being males and 30.4% (14/46) being females. The distribution of underlying diseases among the 46 patients is outlined in Fig 1, with acute leukemia complicated by mucormycosis being the most prevalent, particularly acute myeloid leukemia (AML) and acute lymphoblastic leukemia (ALL) accounting for 26.1% (12/46) and 23.9% (11/46), respectively. The distribution of Mucor strains in patients is presented in Table 1, with *Rhizomucor pusillus* and *Rhizomucor miehei* being the most common, constituting 30.4% (14/46) and 21.7% (10/46), respectively.

All patients exhibited lung involvement, and one case presented with nose-orbit-brain involvement. Table 2 summarizes patient characteristics, prior treatments, and prognostic risk factors. Other comorbidities were observed in 43.5% (20/46) of patients, including hypertension, heart disease, hepatorenal insufficiency, cerebral infarction, splenectomy, and diabetes. Neutropenia was diagnosed in 56.5% (26/46) of patients, with 28.3% (13/46) experiencing neutropenia for more than 10 days. A significant number of patients presented hematological malignancy (80.4%, 37/46) and underwent multiple chemotherapy sessions (80.4%, 37/46). Only one patient did not receive chemotherapy. Additionally, 8.7% (4/46) of patients received rituximab, 28.3% (13/46) underwent hematopoietic stem cell transplantation, 10.9% (5/46) had graft-versus-host disease (GVHD), 8.7% (4/46) had diabetes, and 23.9% (11/46) had CMV infection. Hepatitis virus infection was present in 13.0% (6/46) of patients, human polyomavirus 1 (BK) in 6.5% (3/46), and EBV infection in 6.5% (3/46). A significant portion of patients (84.8%, 39/46) had received antifungal therapy before the diagnosis of mucormycosis infection, and 37% (17/46) had applied high-dose corticosteroids before diagnosis. In this study, the concordance rate between traditional cultivation methods and metagenomic next-generation sequencing (mNGS) detection was found to be 19.57%.

Regarding antifungal therapy after Mucor infection, 60.9% (28/46) of patients received AmB with an efficacy rate of 75%, while 30.4% (14/46) were treated with a conazole with an efficacy rate of 64.3%. Combination therapy involving two

PLOS Neglected Tropical Diseases

**A.**

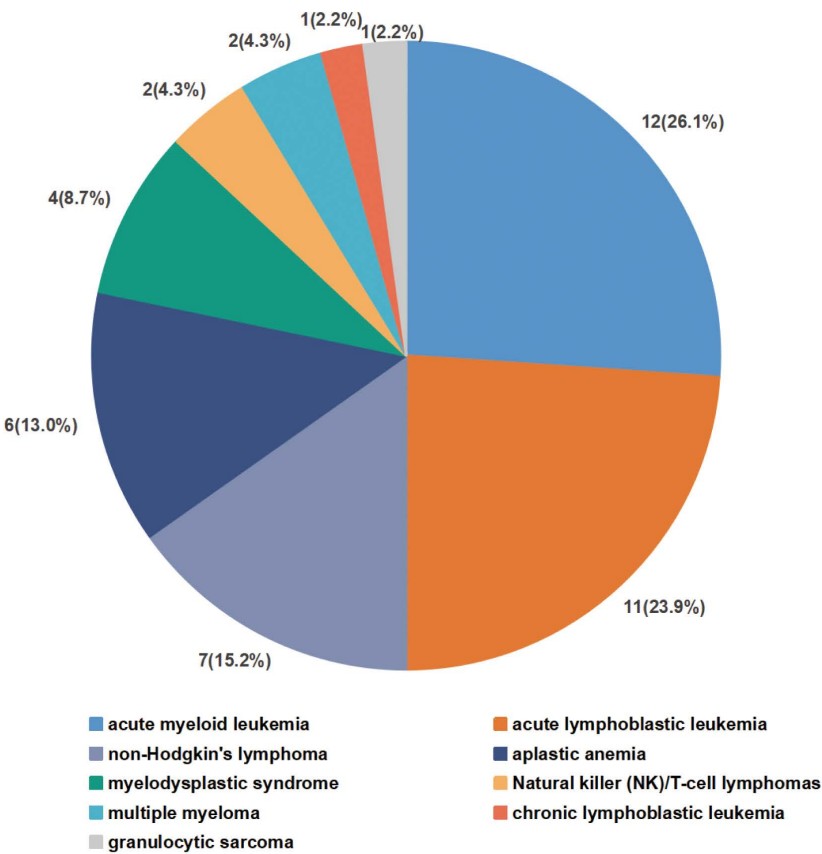

- acute myeloid leukemia
- acute lymphoblastic leukemia
- non-Hodgkin's lymphoma
- aplastic anemia
- myelodysplastic syndrome
- Natural killer (NK)/T-cell lymphomas
- multiple myeloma
- chronic lymphoblastic leukemia
- granulocytic sarcoma

**B.**

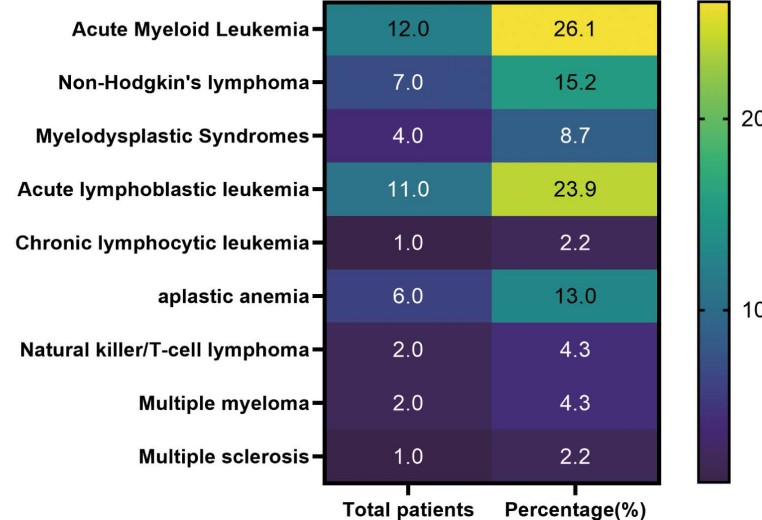

**Fig 1. Classification of hematological diseases in our patient population.**

**Table 1. Mucor strains documented in our patient population.**

| | Total patients | Percentage (%) |
|---|---|---|
| *Rhizomucor pusillus* | 14 | 30.4 |
| *Rhizomucor miehei* | 10 | 21.7 |
| *Rhizopus microspores* | 6 | 13.0 |
| *Rhizopus oryzae* | 4 | 8.7 |
| *Lichtheimia ramosa* | 2 | 4.3 |
| *Lichtheimia corymbifera* | 1 | 2.2 |
| *Cunninghamella bertholletiae* | 1 | 2.2 |
| *Rhizopus microspores* | 1 | 2.2 |
| *Rhizopus delemar* | 1 | 2.2 |
| *Mucor irregularis* | 1 | 2.2 |
| *Rhizomucor miehei* and *Rhizomucor pusillus* | 1 | 2.2 |
| *Rhizopus oryzae*, *Rhizomucor miehei* and *Rhizopus microspores* | 1 | 2.2 |
| *Rhizomucor pusillus* and *Rhizopus microspores* | 1 | 2.2 |
| *Rhizopus microspores* and *Lichtheimia corymbifera* | 1 | 2.2 |
| *Rhizomucor pusillus* and *Cunninghamella bertholletiae* | 1 | 2.2 |
| Total | 46 | 100.0 |

**Table 2. Characteristics, previous treatment, and risk factors of the patient population.**

| Characteristics | | Value (%) |
|---|---|---|
| Median age (range) (years) | | 49(9-78) |
| Age ≥ 60 | | 15(32.6%) |
| Gender | Male | 32(69.6%) |
| | Female | 14(30.4%) |
| Other comorbidities | | 20(43.5%) |
| Neutropenia at diagnosis | | 26(56.5%) |
| Neutrophils decreased for more than 10 days | | 13 (28.3%) |
| Disease state | NR | 37(80.4%) |
| | CR | 9(19.6%) |
| Infection site | Pulmonary type | 48(98.0%) |
| | Lung type and nose-orbit-brain type | 1(2.0%) |
| Chemotherapy sessions | No chemotherapy | 1(2.2%) |
| | single chemotherapy | 8(17.4%) |
| | Multiple chemotherapy | 37(80.4%) |
| Rituximab | | 4(8.7%) |
| Hematopoietic stem cell transplantation | | 13 (28.3%) |
| GVHD | | 5 (10.9%) |
| Diabetes | | 4 (8.7%) |
| CMV infection | | 11 (23.9%) |
| Hepatitis virus | | 6 (13.0%) |
| BK infection | | 3 (6.5%) |
| EBV infection | | 3 (6.5%) |
| Pre-antifungal therapy | | 39(84.8%) |
| high-dose corticosteroids | | 17(37%) |

CR: complete remission

NR: no remission

drugs was administered to 8.7% (4/46) of patients. According to Fisher's exact probability method (p = 0.409 > 0.05), there was no significant difference in the therapeutic effects across the three treatment modalities.

The median OS for all patients in the study was 123 days. Kaplan-Meier survival curves and Log-Rank tests were utilized to evaluate differences in survival. The median OS for patients aged < 60 years was significantly longer than for those aged > 60 years (245 days vs. 164 days, p = 0.004 < 0.05). Patients without other underlying diseases exhibited a significantly longer median OS compared to those with other underlying diseases (69 days vs. 33 days, p = 0.018 < 0.05). Furthermore, patients without high-dose corticosteroids had a significantly longer median OS than those with high-dose corticosteroids (707 days vs. 25 days, p < 0.001). Patients without neutropenia or with neutropenia lasting more than 10 days had a significantly longer median OS than those with neutropenia lasting more than 10 days (382 days vs. 25 days, p = 0.003 < 0.05). Patients with a single Mucor infection had a significantly longer median OS than those with two or more Mucor infections (181 days vs. 14 days, P < 0.001). The Kaplan-Meier survival curve is depicted in Fig 2, and the corresponding hazard ratios (HR) with 95% confidence intervals are calculated by univariate Cox analysis, as shown in Table 3. Variables with p < 0.05 were included in multivariate Cox survival analysis, revealing that age ≥ 60 (p = 0.006 < 0.05), high-dose corticosteroids (p = 0.001 < 0.05), neutropenia lasting more than 10 days (p = 0.041 < 0.05), and two or more Mucor infections (p = 0.004 < 0.05) were independent risk factors for OS in patients with hematological diseases.

## Discussion

Mucormycosis stands as the third most prevalent invasive mycosis among individuals with hematological diseases and those undergoing allogeneic stem cell transplantation, ranking just after candidiasis and aspergillosis. It manifests as a vascular-infiltrating infection caused by filamentous fungi within the Mucor order [9]. The mortality rate associated with mucormycosis ranges widely from 40% to 80%, posing a considerable threat, especially in patients with hematological diseases and those undergoing hematopoietic stem cell transplantation [3,11,14]. Mucormycosis progressed rapidly, and studies have shown that delayed treatment for more than 6 days doubles the mortality rate [15]. Therefore, timely examination, rapid diagnosis, and prompt intervention become crucial when fungal infection is suspected. Early diagnosis of mucormycosis during clinical practice is often challenging, with histopathology and routine culture being the gold standards. Biopsy, particularly in patients with hematological diseases, is intricate. Traditional pathogen culture, due to its time-consuming nature, susceptibility to contamination, and low positive rate, presents limitations [3,4]. In recent years, mNGS has emerged as a culture-independent pathogen identification method, offering rapid, sensitive, and accurate results. It serves as a significant supplement to traditional culture-based methods such as culture [6]. Numerous successful cases and studies have underscored the substantial potential of mNGS in mucormycosis diagnosis [16]. In this study, we enrolled 46 patients with hematological diseases strongly suspected of having mucormycosis. For the 9 patients classified as "Proven", mNGS yielded results before the histopathological examination. Additionally, for "Probable" cases, mNGS proved instrumental in identifying pathogens, a task challenging for traditional methods.

In this study, the concordance rate between traditional cultivation methods and metagenomic next-generation sequencing (mNGS) detection was found to be 19.57%. This finding aligns with previous research [17]. The primary reasons for this discrepancy include the inherent biological characteristics of mucormycosis, which are not conducive to in vitro culture, as well as limitations in sample processing that result in challenges in cultivation and a low positive rate. In contrast, mNGS is a culture-independent approach for the identification of pathogenic microorganisms. It employs high-throughput sequencing technology to detect and characterize microbial DNA and/or RNA in patient samples, subsequently comparing and analyzing these sequences against a comprehensive database. This allows for the unbiased detection of a wide range of infections within the samples [5]. mNGS offers significant advantages, particularly for immunocompromised individuals and patients whose blood culture positivity rates are diminished due to the administration of potent antibiotics. Currently, mNGS is recommended as a first-line diagnostic tool for patients undergoing hematopoietic stem cell transplantation and solid organ transplantation or as an adjunctive diagnostic method [18,19].

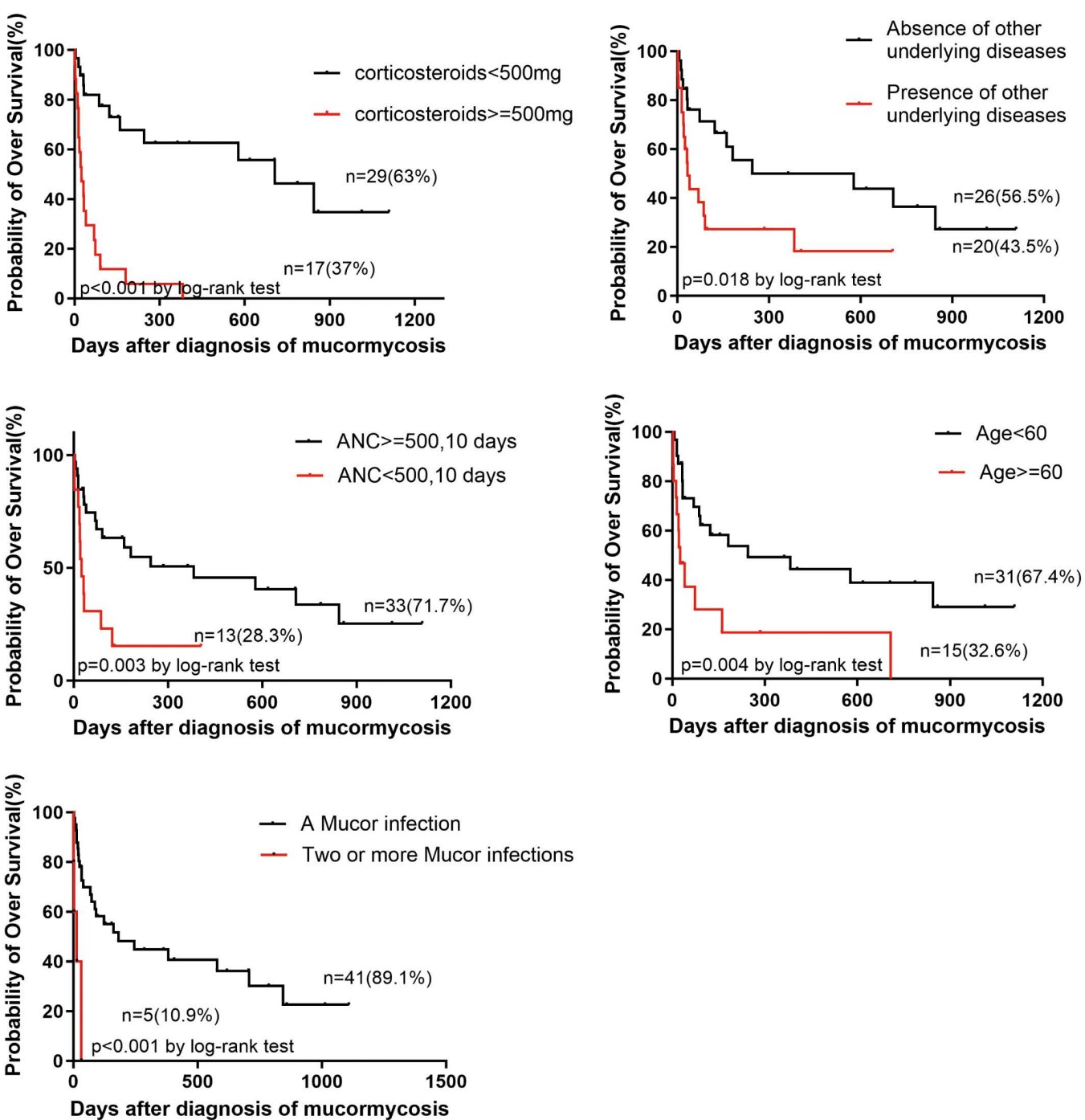

**Fig 2. Kaplan-Meier survival curve illustrating the differences in median OS based on various risk factors.** Patients aged <60 years exhibited a significantly longer median OS compared to patients aged >60 years (245 days vs. 164 days, p = 0.004 < 0.05). Moreover, patients without other underlying diseases had a significantly longer median OS than those with other underlying diseases (69 days vs. 33 days, p = 0.018 < 0.05). Additionally, the median OS of patients without high-dose corticosteroids was significantly longer than that of patients with high-dose corticosteroids (707 days vs. 25 days, p < 0.001). Patients without neutropenia or with neutropenia lasting no more than 10 days exhibited a significantly longer median OS than those with neutropenia lasting more than 10 days (382 days vs. 25 days, p = 0.003 < 0.05). Furthermore, the median OS of patients with one kind of Mucor infection was significantly longer than that of patients with two or more kinds of Mucor infections (181 days vs. 14 days, P < 0.001).

**Table 3. Prognostic factors of mortality in univariate and multivariate analysis.**

| | Univariate analysis P | Univariate analysis HR (95% CI) | Multivariable analysis P | Multivariable analysis HR (95% CI) |
|---|---|---|---|---|
| Age ≥ 60 | 0.004 | 2.894(1.350-6.205) | 0.006 | 3.449(1.438-8.271) |
| Gender | 0.197 | | | |
| Other comorbidies | 0.018 | 2.464(1.137-5.329) | 0.180 | 1.799(0.762-4.246) |
| Neutropenia at diagnosis | 0.523 | | | |
| Primary disease state (CR/ NR) | 0.660 | | | |
| Number of chemotherapy (no chemotherapy/ single chemotherapy/ multiple chemotherapy) | 0.532 | | | |
| Hematological diseases (leukemia/ lymphoma/other) | 0.329 | | | |
| Rituximab | 0.613 | | | |
| Hematopoietic stem cell transplantation | 0.392 | | | |
| GVHD | 0.461 | | | |
| diabetes | 0.782 | | | |
| CMV infection | 0.591 | | | |
| Hepatitis virus | 0.424 | | | |
| EBV infection | 0.828 | | | |
| Pre-antifungal therapy | 0.473 | | | |
| high-dose corticosteroids | <0.001 | 6.800(2.938-15.741) | 0.001 | 5.028(1.990-12.703) |
| BK infection | 0.134 | | | |
| Pre-antifungal therapy | 0.473 | | | |
| Two or more Mucor infections | <0.001 | 7.522(2.507-22.568) | 0.004 | 7.273(1.870-28.287) |
| Fever | 0.184 | | | |
| Three types of treatment (amphotericin B/ triazole/combination of two drugs) | 0.641 | | | |
| Neutrophils decreased for more than 10 days | 0.003 | 3.190(1.434-7.098) | 0.041 | 2.641(1.042-6.693) |

GVHD: graft versus host reaction

CR: complete remission:

NR: no remission

Pulmonary mucormycosis typically results from the inhalation of spores, predominantly affecting patients with hematological diseases and those undergoing hematopoietic stem cell transplantation [11,20]. This condition is characterized by persistent fever (> 38 °C), an enduring cough, dyspnea, chest pain, and hemoptysis. Non-responsive to broad-spectrum antibiotics, the infection often involves vascular infiltration, causing necrosis and leading to the rapid progression of lung disease. In some cases, it can disseminate to other organs, resulting in disseminated mucormycosis, significantly elevating the mortality rate to 95%. The diagnosis of pulmonary mucormycosis is challenging, given the poor specificity of imaging findings. Therefore, rapid diagnosis and prompt treatment intervention are critical for successful management of pulmonary mucormycosis [21]. In this study, all patients had pulmonary Mucor infection, with one case also presenting with nasal-orbital-cerebral Mucor infection. As of September 11, 2023, the mortality rate stood at 63%, aligning with findings from previous studies [9,22].

The treatment approach for mucormycosis involves proactive management of underlying diseases, including controlling elevated blood sugar levels, correcting neutropenia, and reducing the use of immunosuppressive drugs [23]. Regarding drug therapy, it is advisable to initiate empirical therapy in patients with neutropenia or graft-versus-host disease. Surgical treatment should be coupled with continued administration of previously effective antifungal drugs in immunosuppressed

patients with a history of Mucor infection. Immediate treatment is essential for immunodeficient patients suspected of mucormycosis. Currently, monotherapy is commonly employed. First-line monotherapy options include liposomal AmB, isavuconazole, and posaconazole. Liposomal AmB is strongly recommended, and for salvage treatment, the use of two triazole drugs is highly recommended [2,23]. Research has demonstrated that there is no significant difference in the therapeutic efficacy between the use of amphotericin B liposomes alone and their combination with posaconazole [24], aligning with our findings. Conversely, some studies suggest that combining isavuconazole or posaconazole with amphotericin B liposomes is more effective than using amphotericin B liposomes alone [25,26]. Previous research has indicated no significant difference in the therapeutic effects of antifungal drugs for mucormycosis compared to amphotericin B treatment [27], which is consistent with our results. However, other studies have reported that amphotericin B exhibits a stronger therapeutic effect on mucormycosis than combined antifungal drugs [25, 26]. It is important to note that this study is a single-center, small-sample retrospective analysis, which presents certain limitations. Consequently, larger-scale, multi-center studies with more systematic evaluations are necessary in the future.

Currently, immunotherapy stands as a significant research hotspot for the treatment of mucormycosis. Studies have identified Mucor-specific T cells expressing elevated levels of IFN-γ, IL-4, IL-17, and IL-10. T cells sourced from healthy volunteers exhibit effective responses to Mucor, enhancing the killing effect of phagocytes [28,29]. A heightened early Th1 response mediated by higher levels of IFN-γ and IL-2 can confer greater resistance to pulmonary infection, with IL-2 inducing T cell clone expansion [22,30]. Type I IFN-γ and granulocyte-macrophage colony-stimulating factor (GMCSF) can stimulate polymorphonuclear leukocytes, enhancing damage to Rhizopus hyphae [31]. GMCSF or granulocyte infusion has been considered for invasive aspergillosis treatment [32]. There are reports suggesting the effectiveness of Navulizumab combined with interferon γ in treating refractory Mucormycosis, indicating a promising avenue for research on combining immunotherapy with IFN-γ in Mucormycosis treatment [33].

Since 2021, the incidence of COVID-19-associated mucormycosis has been on the rise. Systemic corticosteroid therapy, while reducing mortality in severe COVID-19 cases, increases the risk of secondary mycosis [34]. Corticosteroids are indicated for hospitalized patients with COVID-19 pneumonia who require supplemental oxygen and advanced respiratory support [35]. However, there is a prevalent inappropriate and excessive use of corticosteroids in the management of COVID-19, particularly in outpatient settings [36]. The administration of corticosteroids in COVID-19 cases can elevate the risk of immunosuppression [37], including alterations in macrophage and neutrophil function, insulin resistance, and increased blood sugar [38,39]. Corticosteroids are not considered a major risk factor for Mucor infection in the absence of severe immunosuppression [40]. In an animal study, inhaling Mucor spores in animals with normal immune function did not cause Mucor disease [41]. However, the infection rate and mortality of immunosuppressed animals induced by corticosteroids increased significantly [42]. While maintaining patients on low-dose corticosteroids and ensuring good blood glucose control can help mitigate the risk of mucormycosis, studies have demonstrated that even short-term use of corticosteroids can increase susceptibility to mucormycosis infection [2,43,44]. In this study, we examined immunosuppressed patients with hematological disease and Mucor infection, finding that cumulative prednisolone use exceeding 500mg one month before infection is associated with a statistically significant decrease in overall survival ($p < 0.05$). This finding substantiates that prolonged use of corticosteroids is an independent risk factor for OS, consistent with earlier studies that identified prolonged corticosteroid use (cumulative dose within 4 weeks before infection > 600mg prednisone) as a contributing factor in mucormycosis ($p < 0.05$) [45,46].

In patients with hematological diseases, the role of neutrophils in inhibiting the proliferation of fungal spores is crucial, and the mortality rate for patients with persistent neutropenia is nearly 100%. Pulmonary mucormycosis is most prevalent in leukemia patients undergoing chemotherapy and hematopoietic stem cell transplantation, often accompanied by severe neutropenia [47]. Current evidence suggests that neutropenia can lead to the rapid progression of pulmonary mucormycosis [48]. Studies indicate that neutrophils primarily destroy and may kill Aspergillus fumigatus and Rhizopus oryzae in vitro through an active oxygen-dependent mechanism on the cell surface. Neutrophil cationic proteins damage the hyphae of Rhizopus

oryzae through the myeloperoxidase system, and the effect is enhanced when hydrogen peroxide is used alone at sub-inhibitory concentrations [49]. Both mononuclear and polymorphonuclear phagocytes in normal hosts eliminate Mucor by producing oxidative metabolites and cationic peptide defensins. The lack or impairment of phagocytes has been shown to increase the risk of Mucor infection [39]. A study revealed that neutrophils rapidly induce NF-κB pathway-related genes after exposure to Rhizopus oryzae hyphae, leading to the upregulation of numerous pro-inflammatory genes and Toll-like receptor 2 [50], enhanced antifungal activity [51], In this study, neutropenia for more than 10 days was identified as an independent predictor for OS infection by Mucor in patients with hematological diseases. Previous studies have presented varying conclusions regarding the association between neutropenia and Mucor infection [45]. Whereas many studies confirm the longstanding association between neutropenia and Mucor infection, others add that although poor neutrophil recovery is correlated with Mucor infection, it may not be an independent risk factor for Mucor infection prognosis [52].

This study is a single-center, retrospective case analysis. The sample size is small, and there may be potential selection bias, which brings certain limitations. Therefore, in future research, we plan to conduct a larger-scale, multi-center study.

## Conclusion

In summary, immunocompromised patients with hematological diseases and mucormycosis have rapid infection progression and high mortality. When mucor infection is suspected, early diagnosis and treatment are key to improving prognosis. mNGS may be an important supplementary method for early diagnosis, especially for patients who are not suitable for histopathology or culture. Our study showed that age > 60 years, two or more mucor infections, persistent neutropenia, and large-scale use of corticosteroids are independent risk factors for OS of mucormycosis in patients with hematological diseases. Therefore, the rational use of corticosteroids, minimizing the duration of neutropenia, and enhancing the patient's own immune function are important measures to reduce the patient's mortality.

## Supporting information

**S1 Table. The clinical characteristics, epidemiological features, treatment, and prognosis of 46 patients with hematological diseases and Mucor infection.**
(XLSX)

## Acknowledgments

The authors express gratitude to the patient.

## Author contributions

**Conceptualization:** Yu Cui.

**Data curation:** Yu Cui, Rui Zhao, Ruihua Mi, Lin Chen.

**Formal analysis:** Yu Cui.

**Methodology:** Rui Zhao, Ruihua Mi, Lin Chen.

**Writing – original draft:** Yu Cui.

**Writing – review & editing:** Lin Wang, Dongbei Li, Cheng Cheng, Mengjuan Li, Xudong Wei.

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
