## [Decision Letter · Decision Letter 0]

3 Jul 2025

PNTD-D-25-00614Hematological diseases-Related Mucormycosis: A Retrospective Single Center StudyPLOS Neglected Tropical Diseases Dear Dr. Wei, Thank you for submitting your manuscript to PLOS Neglected Tropical Diseases. After careful consideration, we feel that it has merit but does not fully meet PLOS Neglected Tropical Diseases's publication criteria as it currently stands. Therefore, we invite you to submit a revised version of the manuscript that addresses the points raised during the review process. Please submit your revised manuscript within 30 days Sep 01 2025 11:59PM. If you will need more time than this to complete your revisions, please reply to this message or contact the journal office at plosntds@plos.org. Please include the following items when submitting your revised manuscript: * A rebuttal letter that responds to each point raised by the editor and reviewer(s). You should upload this letter as a separate file labeled 'Response to Reviewers '. This file does not need to include responses to any formatting updates and technical items listed in the 'Journal Requirements' section below.* A marked-up copy of your manuscript that highlights changes made to the original version. You should upload this as a separate file labeled 'Revised Manuscript with Track Changes '.* An unmarked version of your revised paper without tracked changes. You should upload this as a separate file labeled 'Manuscript '.

We look forward to receiving your revised manuscript.

Kind regards,

Ahmed Hassan Fahal, FRCS, FRCSI, FRCSG, MS, MD, FRCP(London)

Academic EditorPLOS Neglected Tropical Diseases

Marcio Rodrigues

Section Editor

Shaden Kamhawi

co-Editor-in-Chief

Paul Brindley

co-Editor-in-Chief

**Journal Requirements:**

1) Please provide an Author Summary. This should appear in your manuscript between the Abstract (if applicable) and the Introduction, and should be 150-200 words long. The aim should be to make your findings accessible to a wide audience that includes both scientists and non-scientists. Sample summaries can be found on our website under Submission Guidelines:

2) Thank you for including an Ethics Statement for your study. Please ensure that it is included under a subheading 'Ethics Statement', at the beginning of your Methods section. 

Note:  The Ethics Statement should include : The full name(s) of the Institutional Review Board(s) or Ethics Committee(s), the approval number(s), or a statement that approval was granted by the named board(s), and a statement that formal consent was obtained (must state whether verbal/written).

4) Tables should not be uploaded as individual files. They should be included in the manuscript. Please remove the separate table files from the online submission form. For more information about how to format tables, see our guidelines:

https://journals.plos.org/plosntds/s/tables 

5) We note that your Data Availability Statement is currently as follows: "All relevant data are within the manuscript and its Supporting Information files." However, there are not any supporting information files uploaded in the submission file inventory.

3) If any authors received a salary from any of your funders, please state which authors and which funders.

7) Please ensure that the funders and grant numbers match between the Financial Disclosure field and the Funding Information tab in your submission form. Note that the funders must be provided in the same order in both places as well. Currently, the order of the grants is different in both places. In addition, this grant "LHGJ20210185" is missing from the Financial Disclosure field.

**Reviewers' comments:**Reviewer's Responses to Questions

**Key Review Criteria Required for Acceptance?**

**Methods**

-Are the objectives of the study clearly articulated with a clear testable hypothesis stated?

-Is the study design appropriate to address the stated objectives?

-Is the population clearly described and appropriate for the hypothesis being tested?

-Is the sample size sufficient to ensure adequate power to address the hypothesis being tested?

-Were correct statistical analysis used to support conclusions?

-Are there concerns about ethical or regulatory requirements being met?

Reviewer #1: Line 69: was culture and nGS performed on all patients included, if not what percentage had either and both?

Additionally, the concordance rate for microbiologic confirmed infection would also be of interest to the reader.

Reviewer #2: Methods appear valid.

Reviewer #3: 1) objectives are clearly stated and the study design matches the objectives

2)population is clearly describes.

3) as far as sample size is concerned, with only 46 patients, subgroup analyses (e.g., multivariate Cox regression for multiple variables) may be underpowered, increasing the risk of type II error. However, given the rarity of hematologic mucormycosis, the sample size is reasonable for a single-center retrospective study, though this limitation should be more explicitly discussed.

4) statistical analyses used is also appropriate

4) yes concerns regarding ethical or regulatory requirements are met

**Results**

-Does the analysis presented match the analysis plan?

-Are the results clearly and completely presented?

-Are the figures (Tables, Images) of sufficient quality for clarity?

Reviewer #1: Given the focus on COVID-19 and the impact on steroid use and thereby assumption of the correlation with Mucor rate of rise, it would be informative to capture the frequency of patient flagged for steroid exposure who also had concomitant COVID-19 diagnosed.

Figure 1 cold benefit from alternative data presentation such as a heatmap

Reviewer #2: Analysis presented matches the analysis plan, and the results are clearly presented.

Reviewer #3: 1) Table 1 (Mucor strain distribution) and Table 2 (patient characteristics) are useful and well-structured, though Table 1 includes redundant rows (e.g., mixed strain listings that are unclear) — this could confuse readers and needs cleanup.

2) Table 3 (Cox analysis) is detailed and informative, but there are formatting inconsistencies — e.g., irregular column alignment and missing confidence intervals in some entries.

3) Consider expanding discussion on treatment efficacy outcomes to provide more clinical interpretation.

**Conclusions**

-Are the conclusions supported by the data presented?

-Are the limitations of analysis clearly described?

-Do the authors discuss how these data can be helpful to advance our understanding of the topic under study?

-Is public health relevance addressed?

Reviewer #1: The authors provide convincing analysis of the regional drivers of Mucor infection in hematologic patients, noting high dose steroid exposure, advanced age, and multiple Mucor infections as clear drivers of increased mortality risk. As noted above, commentary about COVID-19 is made without evidence of infection in captured patient base, this may be related to regional access to testing at this time, but if testing was available as a standard of care, commentary on the association of this diagnosis with steroid exposure in these patients would be of interest to support the complications offered in the conclusion.

Reviewer #2: The conclusions are justified and supported by the data.

Reviewer #3: The discussion lightly touches on some known limitations — e.g., challenges in histopathological diagnosis, limitations of traditional culture methods, and the value of mNGS in such contexts.

However, key limitations are not explicitly acknowledged, such as:

Small sample size (n=46) and its implications for statistical power.

Single-center design and retrospective nature, which may limit generalizability.

Potential selection bias (e.g., inclusion only of patients who underwent mNGS).

**Editorial and Data Presentation Modifications?**

Reviewer #1: I would encourage the authors to define nGS for the reader within the manuscript with first reference as well as in the abstract.

line 319 potentially unintended space between Mucor and mycosis.

Would review Table for unnecessary periods, and consider standard italicized nomenclature in the manuscript where appropriate

Figure 2 grpahic appears blurry in my editorial review sample. Would provide high resolution alternative.

Reviewer #2: (No Response)

Reviewer #3: (No Response)

**Summary and General Comments**

Reviewer #1: The authors provide a regional assessment of Mucor infection in hematologic patients with descriptive statistical approach to the demographics and epidemiology of invasive mucor. The key points made appear supported by the data offered. The manuscript could be strengthened by commentary on concordance rates where standard microbiology techniques and nGS were performed on the same patient to allow for assessment of the non-commercial nGS approach described as it relates to "traditional" mycology workflows.

Reviewer #2: The paper is a retrospective review that describes the clinical characteristics of 46 cases of Mucormycosis diagnosed at the Affiliated Cancer Hospital of Zhengzhou. Using regression analyses, the authors conclude that age greater than 60 years old, prolonged neutropenia (> 10 days), and having two or more Mucor infections were independent risk factors for reduced survival due to Mucormycosis.

While the findings of the paper appear valid, the overall sample size for Mucormycosis is somewhat small and the findings reinforce what is already generally well recognized for infections with fungi in the order Mucorales.

Reviewer #3: (No Response)

PLOS authors have the option to publish the peer review history of their article (what does this mean? ). If published, this will include your full peer review and any attached files.

**Do you want your identity to be public for this peer review?** For information about this choice, including consent withdrawal, please see our Privacy Policy .

Reviewer #1: No

Reviewer #2: No

Reviewer #3: **Yes: ** Syeda Ilsa Aaq

**Figure resubmission:**
---

## [Editor Report · Decision Letter 1]

3 Sep 2025

Dear Doctor Wei,

We are pleased to inform you that your manuscript 'Hematological diseases-Related Mucormycosis: A Retrospective Single Center Study' has been provisionally accepted for publication in PLOS Neglected Tropical Diseases.

Best regards,

Ahmed Hassan Fahal, FRCS, FRCSI, FRCSG, MS, MD, FRCP(London)

Academic Editor

Marcio Rodrigues

Section Editor

Shaden Kamhawi

co-Editor-in-Chief

Paul Brindley

co-Editor-in-Chief

---

## [Editor Report · Acceptance letter]

Dear Doctor Wei,

We are delighted to inform you that your manuscript, "Hematological diseases-Related Mucormycosis: A Retrospective Single Center Study," has been formally accepted for publication in PLOS Neglected Tropical Diseases.

Best regards,

Shaden Kamhawi

co-Editor-in-Chief

Paul Brindley

co-Editor-in-Chief
